# Psychological Distress, Family Support and Employment Status in First-Year University Students in Spain

**DOI:** 10.3390/ijerph16071209

**Published:** 2019-04-04

**Authors:** Jorge Arias-de la Torre, Tania Fernández-Villa, Antonio José Molina, Carmen Amezcua-Prieto, Ramona Mateos, José María Cancela, Miguel Delgado-Rodríguez, Rocío Ortíz-Moncada, Juan Alguacil, Susana Redondo, Inés Gómez-Acebo, María Morales-Suárez-Varela, Gemma Blázquez Abellán, Eladio Jiménez Mejías, Luis Félix Valero, Carlos Ayán, Laura Vilorio-Marqués, Rocío Olmedo-Requena, Vicente Martín

**Affiliations:** 1The Biomedical Research Centre Network for Epidemiology and Public Health (CIBERESP), 28029 Madrid, Spain; carmezcua@ugr.es (C.A.-P.); mdelgado@ujaen.es (M.D.-R.); alguacil@dbasp.uhu.es (J.A.); gomezai@unican.es (I.G.-A.); maria.m.morales@uv.es (M.M.-S.-V.); eladiojimenez@ugr.es (E.J.M.); rocioolmedo@ugr.es (R.O.-R.); vicente.martin@unileon.es (V.M.); 2Agency for Health Quality and Assessment of Catalonia (AQuAS), Carrer de Roc Boronat, 81, 08005 Barcelona, Spain; 3Grupo de Investigación en Interacciones Gen—Ambiente y Salud (GIIGAS)/Instituto de Biomedicina (IBIOMED), Universidad de León, 24071 León, Spain; tferv@unileon.es (T.F.-V.); ajmolt@unileon.es (A.J.M.); 4Department of Preventive Medicine and Public Health, Faculty of Medicine, University of Granada, 18071 Granada, Spain; 5Biosanitary Research Institute of Granada (ibs.GRANADA), University Hospitals of Granada-University of Granada, 18071 Granada, Spain; 6Department of Preventive Medicine and Public Health, University of Salamanca, 37007 Salamanca, Spain; rmateos@usal.es (R.M.); luva@usal.es (L.F.V.); 7Faculty of Education Sciences & Sports, University of Vigo, HealthyFit Research Group Galicia Sur Health Research Institute (IIS Galicia Sur), Sergas-UVIGO, 36005 Pontevedra, Spain; chemacc@uvigo.es (J.M.C.); cayan@uvigo.es (C.A.); 8Division of Preventive Medicine and Public Health, University of Jaén, 23071 Jaén, Spain; 9Departments of Community Nursing, Preventive Medicine and Public Health and History of Science, University of Alicante, 03690 Alicante, Spain; rocio.ortiz@ua.es; 10Research group on Food and Nutrition—Research group of Public Health, University of Alicante, 03690 Alicante, Spain; 11Research Center on Natural Resources, Health, and Environment (RENSMA), University of Huelva, 21071 Huelva, Spain; 12Gerencia de Servicios Sociales, Comisionado Regional para la Droga, 47009 Valladolid, Spain; redmarsu@jcyl.es; 13IDIVAL, University of Cantabria, 39011 Santander, Spain; 14Department of Preventive Medicine and Public Health, Faculty of Pharmacy, University of Valencia, 46100 Burjassot, Spain; 15Department of Biomedical Sciences, Area of Preventive Medicine and Public Health, University of Castilla—La Mancha, 13071 Albacete, Spain; gemma.blazquez@uclm.es; 16Instituto de Investigación Sanitaria del Principado de Asturias (ISPA), 33011 Oviedo, Spain; biolvm00@estudiantes.unileon.es

**Keywords:** psychological distress, family support, employment status, university, social epidemiology, survey study

## Abstract

Mental disorders are consistently and closely related to psychological distress. At the start of the university period, the relationship between a student’s psychological distress, family support, and employment status is not well-known. The aims of this study were: To determine the prevalence of psychological distress in first-year university students and to analyze its relationship with family support and the student’s employment status. Data from 4166 first-year university students from nine universities across Spain were considered. The prevalence of psychological distress was obtained using the GHQ-12, a valid and reliable screening tool to detect poor mental health. To analyze the relationship between psychological distress, family support, and employment status, logistic regression models were fitted. Regarding the prevalence found, 46.9% of men and 54.2% of women had psychological distress. In both genders, psychological distress levels increased as family support decreased. Among women, psychological distress was associated with their employment status. The prevalence of psychological distress among first-year university students in Spain is high. In addition, family support, and employment status for women, could be factors to take into account when developing psychological distress prevention strategies at the beginning of the university period.

## 1. Introduction

The start of university might constitute a crucial period for students’ mental health [1,2,3,4]. During this period, they generally have to face a number of important changes in their lives, like moving, taking on more responsibilities, and adapting to changes in their support network [5]. These changes may create a source of psychological distress, something that has been consistently related to mental diseases. Previous studies have shown that psychological distress might be especially related to mental diseases that have a higher component of neuroticism, like anxiety and depressive disorders [4,6]. Furthermore, it might be particularly relevant in students who have not sufficiently developed appropriate coping strategies to face this transition period, due to the relationship with mental diseases [1,7].

The prevalence of psychological distress found in the university student population is usually lower than in the non-student population, although this varies between studies and may be sensitive to the specific population analyzed and the method of calculation [8,9,10,11]. This prevalence is generally around 20% in the general college student population, but can vary between 5% and 45% depending on the specific population, e.g., medical students or students from a particular country [1,2,4,5,12,13,14,15,16]. Additionally, it is important to highlight the well-documented association of psychological distress and gender in the Spanish context, being higher among women than among men. In terms of socio-economic factors, psychological distress is higher among people with social disadvantages [8,10]. Moreover, starting university could be considered a transition period, with its respective impact on mental health [1]. As such, and in line with previous research [1,2], the prevalence of psychological distress during this transition might be higher than during the rest of the university period and it could be particularly high among students with underdeveloped coping skills.

One important factor for being able to cope with these changes is family support. In line with the concept of emerging adulthood [17,18], students are still free of the obligations inherent to adulthood and they are still dependent on their parents. At the start of university, family support continues to be one of the most relevant sources for students to front their daily obligations, both in an emotional as well as an economic sense. Therefore, a good support system could be an important source of stress relief and a way to facilitate developing adaptive coping mechanisms, but it could also constitute an added stress when support is poor [4,5,7]. Consequently, it is important to examine the relationship between family support and students’ mental health in this period [19].

One socio-economic factor that could be particularly relevant is the student’s employment status, due to its relationship with psychological distress. While this factor has been widely studied in the general population [8,11,20], it has not been studied as much in the university student population. The results of the studies carried out in the general population have consistently shown that being employed could be considered a protective factor against mental health problems [21]. This protective effect in the general population could be related to the lower financial stress that people with a job might have, especially among men. In the university student population, being employed might not be a protective factor, possibly due to the financial pressure that the families of students in this situation might have [13]. In developed countries financial support usually falls on the parents and not on the student. As previous studies consistently show [8,9,10], economic disadvantages are a risk factor to developing psychological distress. In addition, having a job could constitute a source of stress for students and might lead to academic burnout, due to the overload that paid work could add to studying [22,23]. This overload could be related to psychological distress and mental health problems to a greater extent in women than in men [13]. Therefore, unlike the general population, we hypothesize that being employed might be a risk factor for psychological distress in the university student population and that this relationship could be different depending on gender.

In this framework, the aims of this study are: to determine the prevalence of psychological distress among first-year university students in Spain by gender, socio-demographic characteristics and self-perceived health status; and to analyze how psychological distress is related to family support and students’ employment status in this population.

## 2. Materials and Methods

### 2.1. Design, Participants and Procedure

A cross-sectional design was performed, based on the uniHcos Project data [24]. The study population consisted of 4166 first-year university students from 9 Spanish universities (Universities of: León, Cantabria, Jaén, Vigo, Granada, Huelva, Salamanca, Valladolid y Alicante) who signed an informed consent and completed the uniHcos questionnaire between October 2011 and March 2015. All students from these universities were invited to participate, regardless of their field of study. The university degree with the highest representation in the sample was nursing (11% of the total sample), followed by psychology (6%) and law (4%). The uniHcos questionnaire is an online self-administered survey that contains information regarding socio-economic and health related variables, as well as mental health. The latter is assessed through the 12-item version of the General Health Questionnaire (GHQ-12), a screening tool used to detect possible mental health problems [25,26]. The questionnaire was sent to the students’ university email address after their involvement in the project was approved by the university’s respective ethics committee. The overall response rate was 4.6%.

All ethics committees of the collaborating universities evaluated and accepted the project and the participants collaborated voluntarily without compensation. Each participant completed a written informed consent online accepting the conditions of the study. The SphinxOnline® (Le Sphinx: Chavanod, France) platform used in this study kept data confidential and thus complied with the regulations of the 15/1999 Data Protection Act, creating two independent codified databases (one with personal data and one with the questionnaire data). This way, no researcher knew who corresponded to which questionnaire.

Due to possible differences within the study population, the following participants were excluded: Students who were over 25 years old (*n* = 314; 7.5%); those who did not respond to their degree choice (*n* = 4; 0.1%); those that affirmed that they were living with their children (*n* = 13; 0.3%); those for whom sexual orientation was undetermined (*n* = 41; 1.1%), and students living in their own home or whose living quarters were undetermined (*n* = 86; 2.3%). A participant could have missing values in different variables. In total, a sample of 3717 students with a mean age (standard deviation) of 19 (2) years, both in men and women, was considered for the analysis.

### 2.2. Study Variables

#### 2.2.1. Psychological Distress

Psychological distress was considered as the main outcome and assessed using the General Health Questionnaire (GHQ-12), a valid and reliable screening tool to detect poor mental health. The GHQ-12 evaluates the subjective mental state of the person in the non-psychiatric, general population. It has shown suitable psychometric properties for its use in the Spanish population, with an acceptable internal consistency (Cronbach α of 0.75) and an Area Under the Response Operative Curve (AUC) of 0.8 [25,26]. The questionnaire consists of 12 Likert-type items with a 4-point response scale. A 2-point scoring method was used, assigning 0 points to answers 0 and 1, and 1 point to answers 2 and 3. The scores from the 12 items were then added together, obtaining a total score between 0 and 12. A score of 3 or greater was considered psychological distress as suggested by the authors as well as previous research that used this tool [8,9,27].

#### 2.2.2. Family Support and Student Employment Status

These factors were considered as the main explanatory variables of the study.

The student’s family support was considered and assessed using the Family APGAR, an instrument used to determine the level of family support, i.e. to see if the family can be considered as a resource for its members or if it will negatively affect their situation. It has shown sufficient properties to assess the perceived general state of the family’s support at a given time [28], showing acceptable internal consistency (Cronbach α = 0.84) when tested in the Spanish population [29]. The questionnaire was developed in 1978 by Smilkstein et al. [28], and is composed of 5 Likert-type items with a 3-point scale from 0 to 2 and is still in use in Spain in various contexts [30,31,32]. The scores were categorized according to the authors’ recommendations: 7–10 points, normal support; 3–6 points, slightly dysfunctional; and 0–2 points, severely dysfunctional.

The student employment status was considered as a categorical variable with the following levels: only studying, studying, and looking for work, studying, and currently working.

#### 2.2.3. Oher Socio-Demographic Factors and Self-Perceived Health

The following socio-demographic variables were considered: the students’ age (in years, considered as a continuous variable); the university that the student attends (University of Alicante, University of Cantabria, University of Granada, University of Huelva, University of Jaén, University of León, University of Salamanca, University of Valladolid and University of Vigo); whether the degree selected by the student was their first choice (yes, no); the student’s sexual orientation (heterosexual, bisexual or homosexual); place of residence in relation to the parents’ home (same town, same province, same autonomous community (AC), other AC, or other country); the residence in which the student was living during the school year (student residence hall, rented apartment, family household) and the people the student lived with (parents, roommates, significant other, alone).

Self-perceived health status was evaluated with a 5-point Likert-type scale in which the students had to indicate their health status over the last 12 months, ranging from 1 (very good) to 5 (very poor). Due to limitations related to the number of students in some of the categories, the variable was dichotomized into: good health status (scores 1 and 2) and poor health status (scores 3, 4 or 5).

### 2.3. Data Analysis

A descriptive analysis of the population’s characteristics and psychological distress prevalence was carried out. Furthermore, differences by sex and other independent variables were evaluated. To assess possible differences, both in the study variables as well as in the distribution of the prevalence through these variables, a chi-square test for categorical variables was used and a Mann-Whitney U test for age, due to its non-normal distribution. To identify factors related to psychological distress, a bivariate and a multivariate analysis were performed through logistic regression models. From these, crude Odds Ratios (OR), adjusted Odds Ratios (aOR) and their respective 95% Confidence Intervals (95%CI) were calculated. All models were stratified by sex. Multivariate models were adjusted for family support (Family APGAR), university, first choice when selecting a degree, sexual orientation, who the student lived with, working status, and self-perceived health status. The final selection of variables included in the multivariate models was based on the proposed framework and used a backward stepwise method with a significance level of 0.1 for removal from the model and 0.05 for addition to the model. The goodness of fit of multivariate models was evaluated using the Hosmer and Lemeshow test and the absence of multi-collinearity was verified. The statistical significance level was fixed at 95% (α = 0.05). All analyses were carried out using the statistical software Stata v.14 (StataCorp LLC: College Station, TX, USA) [33].

### 2.4. Ethics Approval and Consent to Participate

The ethics committees of the collaborating universities evaluated and accepted this procedure (Ethics Committee of University of Granada, Ethics Committee of University of Huelva, Ethics Committee of University of Jaén, Ethics Committee of University of León, Ethics Committee of University of Salamanca and Ethics Committee of University of Vigo) and the participants collaborated voluntarily without compensation.

## 3. Results

Table 1 shows differences between men and women in the characteristics of the analysed sample. While men mostly lived in the same town (30.6%), women mostly lived in the same province (34.4%) as their parents. Statistically significant differences (*p* < 0.05) between men and women were only found in sexual orientation, people they lived with and self-perceived health status. Differences between men and women in family support and employment status were not found.

As for the prevalence of psychological distress (Table 2), the overall prevalence was lower in men (46.9%) than in women (54.2%). Furthermore, a higher prevalence among women was found in all categories of all the variables with the only exception being in poor self-perceived health status where the prevalence was slightly higher among men (71.2%) than among women (71.1%). Regarding prevalence differences across the specific variables at the bivariate level, statistically significant differences related to family support and self-perceived health status were found in both sexes. Only among women were differences in psychological distress found to be related to employment status, sexual orientation, and the people the student lived with.

Taking into account the results of the multivariate analysis (Table 3), similarities and differences by sex in factors related to psychological distress were found. In both sexes family support was closely related to psychological distress. As levels of family support decrease, psychological distress increases. Furthermore, employment status among women was associated with psychological distress, with a higher prevalence of distress observed in women who were studying and looking for work (OR: 1.32; 95%CI: 1.08–1.61) compared to those that were only studying.

## 4. Discussion

The results of the present study place the prevalence of psychological distress among first-year university students in Spain at around 50%. This prevalence is higher than what previous studies have obtained with this population and what has sometimes been observed in other studies [2,4,5,12,13,15]. For this reason, we consider it necessary to comment upon these differences due to their possible implications for preventing the development of mental health illnesses in this population.

The obtained results show a higher prevalence than previous studies carried out in the university student population [2,4,5,15], with the exception of two studies. One study carried out in an Italian university student population had a prevalence of about 50% [13], while the other, in a UK student population, had a prevalence of about 40% [1]. This similarity in the prevalence found in Italy, UK and Spain, could be due to sameness in the context and culture of the European countries and to differences in the tool used for the assessment of the prevalence. Therefore, and due to the high prevalence of psychological distress found, we consider adequate to longitudinally and continuously monitor it in this first-year university context, especially in these countries, and our results could serve as baseline for this monitoring.

The fact that our results indicate a prevalence nearly twice that found previously might be explained by the discrepancy between the characteristics of the populations analyzed in the different studies. Most of these studies did not focus on the first-year university student population. As suggested by previous research [1,4,7,17,18,19], first-year university students could be affected by more pressure or stress than upper classmen due to the transition period with many changes in various aspects of the students’ lives. This aspect of stress could constitute a risk factor for developing physical and mental health problems, something that has been proven previously [4,34,35]. As such, future research should take into account if students are starting university, their transition to university, and their mechanisms for coping with aspects of university, beginning at the start of the university period, as well as their adjustment transitioning to university and their stress coping strategies [7,17]. These factors may help more precisely quantify and assess whether the first year of university study is or is not a key period in preventing mental health problems.

There was a clear gradient found between an increase in psychological distress and worse family support. The causal relationship between these variables could be bi-directional, meaning dysfunctional family support may be the cause of psychological distress and vice-versa. As such, no causal implications can be drawn from the study, however, we feel it is important to include this factor in the study because of the implications that it could have in the prevention of posterior mental health problems. As shown by previous research, focused both in psychological distress and in academic burnout, [4,5,22,23], family support might be a way to alleviate stress for students. In addition, if there was sufficient family support, this could help the individual develop their own healthy and adapted coping skills. Accordingly, we propose that providing the students with strategies to improve family support (e.g., implementing programs to facilitate open communication within the household) could considerably reduce the impact that the beginning of university has on students’ mental health [14,36,37].

Considering students’ employment status, there was a relationship between psychological distress and studying, and looking for work. This relationship, could be explained by the higher financial pressure that the individuals in this situation could have [10,34,38]. Financial pressure may constitute an additional source of stress on top of that provoked by the changes at the start of university. This stress overload could lead to increased psychological distress levels among individuals in this situation. Furthermore, it is possible that the students in this situation might be forced to go from adolescence to adulthood without going through emerging adulthood due to financial pressure [17,34,38]. This abrupt change, could also be related to psychological distress, something that should be considered in future studies. However, the association between psychological distress and simultaneously studying and looking for work was only found among women, which future research should consider and explore.

There are a few limitations to the study, the first being design. Its cross-sectional nature precludes causal interpretation. However, since this study is the starting point for further analyses with a longitudinal perspective, we consider the design to be appropriate to meet our objectives. Secondly, the sample, the response rate and their implications for interpreting the results. We use data derived from the uniHcos project, whose objective is the creation of a cohort rather than to generate data for use in cross-sectional studies. Despite the low response rate caused by the voluntary participation and the possible self-selection bias, this sample could be considered sufficient to achieve the aims of the present study, given its size, the heterogeneity of the degrees in which the students were enrolled and the similarities found in the students’ characteristics between universities. Thirdly, the questionnaire used to determine the prevalence is a screening tool, which could overestimate the prevalence and in addition, there was observe an absence of the same instruments to compare cross-countries data. However, previous studies have shown that family APGAR has adequate psychometric properties meaning that any possible overestimation could be considered acceptable in our context and would not explain the higher prevalence found [25,26]. Finally, it is important to address the question of bias. One potentially important source of bias is related to data collection. As previously discussed, the start of university is a time with many important changes and students may be exposed to different stresses. Consequently, it is possible that we overestimate the prevalence of psychological distress. Despite this, we feel that it is pertinent to consider the prevalence at this time because of the implications on developing mental health illnesses and their prevention.

## 5. Conclusions

The prevalence of psychological distress among first-year university students is much higher than that observed in previous studies in the general university student population, meaning that the start of university could be a key time to monitor psychological distress and prevent its possible consequences. Additionally, our results suggest that both family support and student’s employment status could be particularly relevant factors in developing prevention strategies against the onset of mental health diseases in this specific population.

## Figures and Tables

**Table 1 ijerph-16-01209-t001:** General characteristics of the study population and differences by gender. UniHcos project 2016.

	Men		Women		
	(*n* = 1025)		(*n* = 2692)		
	*n*	%	*n*	%	*p*-Value
**Family support**					0.123
Normal	732	71.41	1993	74.03	
Slightly dysfunctional	222	21.66	503	18.68	
Severely dysfunctional	71	6.93	196	7.28	
**Employment status**					0.090
Only studying	739	72.10	1935	71.88	
Studying and looking for work	211	20.59	606	22.51	
Studying and currently working	75	7.32	151	5.61	
**University**					0.370
University of Alicante	46	4.49	136	5.05	
University of Cantabria	23	2.24	54	2.01	
University of Granada	365	35.61	958	35.59	
University of Huelva	41	4.00	141	5.24	
University of Jaén	53	5.17	161	5.98	
University of León	120	11.71	355	13.19	
University of Salamanca	179	17.46	406	15.08	
University of Valladolid	47	4.59	117	4.35	
University of Vigo	151	14.73	364	13.52	
**First choice when selecting a degree**					0.113
Yes	816	79.61	2078	77.19	
No	209	20.39	614	22.81	
**Sexual orientation**					<0.001
Heterosexual	891	86.93	2491	92.53	
Homosexual	85	8.29	65	2.41	
Bisexual	49	4.78	136	5.05	
**Place of residence**					<0.001
Same town	314	30.63	688	25.56	
Same province	309	30.15	926	34.40	
Same autonomous community (CA)	261	25.46	604	22.44	
Other CA or country	141	13.76	474	17.61	
**Residence**					0.211
Family household	496	48.39	1216	45.17	
Student residence hall	118	11.51	333	12.37	
Rented apartment	411	40.10	1143	42.46	
**People the student lives with**					0.014
Parents	494	48.20	1225	45.51	
Roommates	417	40.68	1209	44.91	
Significant other	21	2.05	74	2.75	
Alone	93	9.07	184	6.84	
**Self-perceived health status**					<0.001
Good	872	85.07	2084	77.41	
Poor	153	14.93	608	22.59	
Mean age (*SD*)	19 (2)		19 (2)		0.072

Note: *p*-value: chi square test for categorical variables and Mann-Whitney U test for continuous variables. SD: standard deviation.

**Table 2 ijerph-16-01209-t002:** Distribution of the prevalence of psychological distress in terms of population characteristics by gender. Bivariate analysis. UniHcos project 2016.

	Men				Women			
	(*n* = 1025)				(*n* = 2692)			
	*n*	%	OR (95% CI)	*p*	*n*	%	OR (95% CI)	*p*
**Total**	481	46.93			1460	54.23		
**Family support**								
Normal	287	39.21	1.00		979	49.12	1.00	
Slightly dysfunctional	144	64.86	2.86 (2.09–3.91)	<0.001	327	65.01	1.92 (1.57–2.36)	<0.001
Severely dysfunctional	50	70.42	3.69 (2.17–6.28)	<0.001	154	78.57	3.80 (2.67–5.40)	<0.001
**Working status**								
Only studying	335	45.33	1.00		998	51.58	1.00	
Studying and looking for work	111	52.61	1.34 (0.99–1.82)	0.062	377	62.21	1.54 (1.28–1.86)	<0.001
Studying and currently working	35	46.67	1.05 (0.66–1.70)	0.825	85	56.29	1.21 (0.87–1.69)	0.265
**University**								
Alicante	184	50.41	1.00		517	53.97	1.00	
Cantabria	19	41.30	0.69 (0.37–1.29)	0.246	66	48.53	0.80 (0.56–1.15)	0.235
Granada	8	34.78	0.52 (0.22–1.27)	0.152	26	48.15	0.79 (0.46–1.37)	0.405
Huelva	21	51.22	1.03 (0.54–1.97)	0.922	82	58.16	1.19 (0.83–1.70)	0.351
Jaén	24	45.28	0.81 (0.46–1.45)	0.486	91	56.52	1.11 (0.79–1.55)	0.547
León	49	40.83	0.68 (0.45–1.03)	0.069	197	55.49	1.06 (0.83–1.36)	0.622
Salamanca	77	43.02	0.74 (0.52–1.06)	0.105	222	54.68	1.02 (0.82–1.30)	0.809
Valladolid	20	42.55	0.72 (0.39–1.35)	0.312	58	49.57	0.84 (0.57–1.23)	0.369
Vigo	79	52.32	1.08 (0.74–1.58)	0.693	201	55.22	1.05 (0.83–1.34)	0.683
**First option in the moment of select degree**								
Yes	374	45.83	1.00		1101	52.98	1.00	
No	107	51.20	1.24 (0.91–1.68)	0.166	359	58.47	1.25 (1.04–1.50)	0.017
**Sexual orientation**								
Heterosexual	408	45.79	1.00		1333	53.51	1.00	
Homosexual	45	52.94	1.33 (0.85–2.08)	0.208	40	61.54	1.39 (0.84–2.31)	0.202
Bisexual	28	57.14	1.58 (0.88–2.82)	0.124	87	63.97	1.54 (1.08–2.21)	0.018
**Place of residence**								
Same town	142	45.22	1.00		372	54.07	1.00	
Same province	142	45.95	1.02 (0.75–1.41)	0.855	507	54.75	1.03 (0.84–1.25)	0.786
Same autonomous community (CA)	123	47.13	1.08 (0.78–1.50)	0.648	331	54.80	1.03 (0.83–1.28)	0.792
Other CA or country	74	52.48	1.34 (0.90–1.99)	0.152	250	52.74	0.95 (0.75–1.20)	0.656
**Residence**								
Family household	228	45.97	1.00		649	53.37	1.00	
Student’s residence hall	50	42.37	0.86 (0.58–1.30)	0.481	181	54.35	1.04 (0.82–1.33)	0.750
Rented apartment	203	49.39	1.15 (0.88–1.49)	0.304	630	55.12	1.07 (0.91–1.26)	0.395
**People that the student lives with**								
Parents	225	45.55	1.00		649	52.98	1.00	
Roommates	207	49.64	1.18 (0.91–1.53)	0.218	652	53.93	1.04 (0.89–1.22)	0.639
Significant other	10	47.62	1.09 (0.45–2.61)	0.852	51	68.92	1.97 (1.19–3.26)	0.009
Alone	39	41.94	0.86 (0.55–1.35)	0.521	108	58.70	1.26 (0.92–1.73)	0.148
**Self–perceived health status**								
Good	372	42.66	1.00		1028	49.33	1.00	
Poor	109	71.24	3.33 (2.29–4.84)	<0.001	432	71.05	2.52 (2.07–3.06)	<0.001
Age			1.05 (0.98–1.12)	0.144			1.03 (0.99–1.07)	0.191

Note: *n*: number of students with psychological distress. %: prevalence of psychological distress. OR: crude Odds Ratio; 95% CI: 95% Confidence Interval; *p*: *p* value from Wald tests.

**Table 3 ijerph-16-01209-t003:** Relationship between family support (APGAR) and working status with psychological distress by gender. Multivariate analysis. UniHcos project 2016.

	Men (*n* = 1025)		Women (*n* = 2692)	
	aOR (95% CI)	*p*	aOR (95% CI)	*p*
Family support				
Normal	1.00		1.00	
Slightly dysfunctional	1.78 (1.44–2.19)	<0.001	1.78 (1.44–2.19)	<0.001
Severely dysfunctional	3.34 (2.33–4.80)	<0.001	3.31 (2.30–4.75)	<0.001
Working status				
Only studying	1.00		1.00	
Studying and looking for work	1.16 (0.64–1.81)	0.354	1.32 (1.08–1.61)	0.006
Studying and currently working	0.90 (0.54–1.50)	0.688	1.06 (0.74–1.50)	0.758

Hosmer and Lemeshow test >0.1 for all models; aOR: adjusted Odds Ratio. Adjusted by family support, university, first choice in degree selection, sexual orientation, who the student lives with, working status, and self-perceived health status. Variables were selected using a backward stepwise method (0.1 for removal from the model and 0.05 for addition to the model). 95%CI: 95% Confidence Interval. *p*: *p*-value.

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
