# Peer review of "Psychological Distress, Family Support and Employment Status in First-Year University Students in Spain"

_ijerph, 2019, doi:10.3390/ijerph16071209_

Round 1

Reviewer 1 Report

The manuscript analyzes the predictive capacity of family support and employment status on stress in first-year university students. As strengths to be highlighted, the work is clear in its objectives and these are coherent with the design developed. It is also worth mentioning the size of the sample used.

However, the work also presents several weaknesses that, in my opinion, should be addressed by the authors in order to recommend their publication. Here are the aspects to be reviewed:

Title:

Since the authors focus exclusively on first-year students, this must be specified in the title of the work. In this regard, according to what is indicated later, the first university course has some peculiar connotations that make it different from the later courses.

Introduction:

In general, I think that this section needs more elaboration. Specifically, it is necessary to clarify the novelty and contribution of this study with respect to previous research. A simple review of the papers in the field allows us to appreciate that both family support and employment status have previously been linked to stress in university students.

For example:

Brougham, R. B., Zail, C. M., Mendoza, C. M., & Miller, J. R. (2009). Stress, sex differences, and coping strategies among college students. Current Psychology, 28, 85-97. doi:10.1007/s12144-009-9047-0

Friedlander, L. J., Reid, G. J., Shupak, N., & Cribbie, R. (2007). Social support, self-esteem, and stress as predictors of adjustment to university among first-year undergraduates. Journal of College Student Development, 48(3), 259-274.

Other major questions that should be addressed:

1) In line 55, it is indicated that stress has been consistently associated with mental diseases. This term is too generic, so it should be specified what consequences have university stress on the students´ physical and psychological health (what are the most frequent clinical manifestations of stress in the first year of university?). There is a lot of prior research on this, so it seems reasonable that the authors delve into this question.

2) Lines 57-62: It seems that the term family support is limited here to that of financial support. As a result, another of the functions of family support, such as emotional support, is omitted. If the authors are limited to family financial support in this study, they should specify it.

3) Line 60: “Thus, family support could constitute an important source of stress discharge when this support is good but could also constitute an extra source of stress when it is poor”. Is there previous empirical evidence for this?

4) Lines 63-70: This paragraph should be located before the previous one. Likewise, since this study focuses on students in the first year of university (according to the authors, this course is especially vulnerable to stress), it should be deepened to a greater degree in the characteristics of the start of university as well as the evolutionary characteristics of people who usually start at university (young adults). Some data about the prevalence of stress in the first year should be considered.

5) Lines 73-75: “The results of the studies carried out in the general population have consistently shown that have employment could be considered as a protective factor against mental health problems [19]”. Why?

6) What hypothesis do the authors establish regarding the impact of family support and employment status on stress according to gender? This hypothesis must be conveniently justified.

Methods:

Design, participants and procedure:

Specify characteristics of participants such as average age and standard deviation. It could also be relevant to know what degrees of each University participated in the study (there may be large differences in stress among degrees).

Study variables:

FAMILY APGAR: What is the author or authors of the instrument? What variables of family support does the instrument contemplate? What are the psychometric properties of the questionnaire?

I have some doubts about the convenience of dichotomizing the variable health status, given that it leads to a certain loss of information.

Data analysis:

Lines 158-170: All this information is left over

Results:

There is a series of variables (sexual orientation, people with live and self-perceived health status) that differentially predict stress in women and men. But it is not clear if these variables have been included in the equation when analyzing the impact of family support and employment status. In other words, it would be convenient (if it has not been done) to analyze their interaction (i.e., dummy variables) with family support and employment status.

It is determined which variables produce stress, but the influence of each observation on the prediction is not quantified (influence measures).

Some figures are not expressed according to APA standards (decimal separated by a point, not by comma).

Discussion:

Lines 235-239: “Accordingly, we propose that providing the students with strategies that improve family support (e.g. implementing programs to facilitate an open communication style inside the household) could considerably reduce the impact that the beginning of the university period has on the mental health of the students [15,28,29]”. As far as I understand, in this study, only financial family support has been analyzed, so the suggestions should accommodate this question.

In general, the implications of the results should be analyzed in greater depth.

Likewise, more study limitations should be indicated (e.g., cross-sectional study, use of self-report procedures, etc.).

Author Response

First of all, we would like to thank the reviewers for the effort and time employed in reviewing our article. We found the comments and suggestions pertinent and are convinced they have helped us improve our paper considerably.

We have responded to all of the comments one by one here below, and have adjusted the main document accordingly.

We hope these changes meet your expectancies and we would like to thank you again for the effort and time spent reviewing our article.

Title:

Since the authors focus exclusively on first-year students, this must be specified in the title of the work. In this regard, according to what is indicated later, the first university course has some peculiar connotations that make it different from the later courses.

RE: We completely agree with the reviewer in the peculiarity of the population analysed. As such, we have reformulated the title.

Changes in text:

Please see title

Introduction:

In general, I think that this section needs more elaboration. Specifically, it is necessary to clarify the novelty and contribution of this study with respect to previous research. A simple review of the papers in the field allows us to appreciate that both family support and employment status have previously been linked to stress in university students.

For example:

Brougham, R. B., Zail, C. M., Mendoza, C. M., & Miller, J. R. (2009). Stress, sex differences, and coping strategies among college students. Current Psychology, 28, 85-97. doi:10.1007/s12144-009-9047-0

Friedlander, L. J., Reid, G. J., Shupak, N., & Cribbie, R. (2007). Social support, self-esteem, and stress as predictors of adjustment to university among first-year undergraduates. Journal of College Student Development, 48(3), 259-274.

RE: We greatly appreciate this comment and we have now elaborated in a greater extent the introduction. In this sense, we have clarified the novelty of the contribution of our study and extended the whole introduction to give a wider focus of our work. Also, we have added some new references.

Changes in text:

Please see highlighted changes in the introduction and references

Other major questions that should be addressed:

1) In line 55, it is indicated that stress has been consistently associated with mental diseases. This term is too generic, so it should be specified what consequences have university stress on the students´ physical and psychological health (what are the most frequent clinical manifestations of stress in the first year of university?). There is a lot of prior research on this, so it seems reasonable that the authors delve into this question.

RE: We completely agree with the reviewer about this specification. In response, we have specified what consequences have university stress on the students´ physical and psychological health.

Changes in text:

Please see lines 55 to 58 of the introduction section

2) Lines 57-62: It seems that the term family support is limited here to that of financial support. As a result, another of the functions of family support, such as emotional support, is omitted. If the authors are limited to family financial support in this study, they should specify it.

RE: We agree with this comment and thus more information regarding the family support is now provided. It should be noted that, as we have clarified in text, the support referred in this paragraph not only includes financial support, also includes emotional support. We apologize for the misunderstanding.

Changes in text:

Introduction, please see 74 to 80

3) Line 60: “Thus, family support could constitute an important source of stress discharge when this support is good but could also constitute an extra source of stress when it is poor”. Is there previous empirical evidence for this?

RE: We have now included a reference in this sense that could serve to support this affirmation.

Changes in text:

Introduction, please see line 80

Please see references

4) Lines 63-70: This paragraph should be located before the previous one. Likewise, since this study focuses on students in the first year of university (according to the authors, this course is especially vulnerable to stress), it should be deepened to a greater degree in the characteristics of the start of university as well as the evolutionary characteristics of people who usually start at university (young adults). Some data about the prevalence of stress in the first year should be considered.

Re: We agree with this suggestion and thus we have relocated the paragraph. In addition, we have deepened to a greater extent in the characteristics of the start of university period as well as the characteristics of young adults that make them vulnerable to psychological distress.

Changes in text:

Introduction, please see lines 65 to 72

5) Lines 73-75: “The results of the studies carried out in the general population have consistently shown that have employment could be considered as a protective factor against mental health problems [19]”. Why?

Re: We completely agree with this comment and therefore, we have added some reasons about why to have an employment could be considered as a protective factor for the general population and a risk factor for the students.

Changes in text:

Introduction, please see lines 87 to 95

6) What hypothesis do the authors establish regarding the impact of family support and employment status on stress according to gender? This hypothesis must be conveniently justified.

RE: A better specification of the differences by gender in the relationship of the employment status and the psychological distress was added.

Changes in text:

Introduction, please see lines 87 and 98

Methods:

Design, participants and procedure:

Specify characteristics of participants such as average age and standard deviation. It could also be relevant to know what degrees of each University participated in the study (there may be large differences in stress among degrees).

RE: Average age of participants and its respective standard deviation was added in this section. In addition, a sentence specifying participants from all university degrees and the three most commonly represented was included.

Changes in text:

Methods, please see lines 109 to 111 and 124 to 130

Study variables:

FAMILY APGAR: What is the author or authors of the instrument? What variables of family support does the instrument contemplate? What are the psychometric properties of the questionnaire?

RE: A better explanation of the development and measurement properties of the Family APGAR was included.

Changes in text:

Methods, please see lines 146 to 151

I have some doubts about the convenience of dichotomizing the variable health status, given that it leads to a certain loss of information.

RE: We acknowledge this appreciation. In this sense, we agree with the comment that we can loss information. Nevertheless, the objective of include this variable is mainly to adjust the analyses for it. Due to this, and after cautiously consider this comment, we have decided to maintain this variable as a dichotomous one. In addition, some categories had a low number of individuals, reason for what to pool their punctuations could provide more robust measures.

Changes in text:

Methods, please see line 170 and 171

Data analysis:

Lines 158-170: All this information is left over

RE: We apologize for the mistake. Consequently, we have taken out these lines.

Changes in text:

Methods, please see data analysis section

Results:

There is a series of variables (sexual orientation, people with live and self-perceived health status) that differentially predict stress in women and men. But it is not clear if these variables have been included in the equation when analyzing the impact of family support and employment status. In other words, it would be convenient (if it has not been done) to analyze their interaction (i.e., dummy variables) with family support and employment status.

RE: We appreciate very much this comment. In this sense, we have added in the statistical analysis section the method used for the selection of the study variables finally included in the multivariable models. Also all models were stratified by gender based on the framework proposed. Due to this, the gender effect could be considered as controlled to a certain extent.

Changes in text:

Methods, please see data analysis section.

It is determined which variables produce stress, but the influence of each observation on the prediction is not quantified (influence measures).

RE: We really appreciate this comment, which we consider completely appropriate. After careful consideration and review of the literature on this topic, we have decided not to change the analyses in this sense. Related to this, we should highlight that, with our relatively large sample size and the distribution of the sample within the levels of the variables included, the results obtained could be considered as robust, as can be seen in the confidence intervals of regression models, and the influence of an individual observation on predictions could not be important.

Some figures are not expressed according to APA standards (decimal separated by a point, not by comma).

RE: We apologize for the mistake and we have corrected it.

Changes in text:

Results, please see text and tables

Discussion:

Lines 235-239: “Accordingly, we propose that providing the students with strategies that improve family support (e.g. implementing programs to facilitate an open communication style inside the household) could considerably reduce the impact that the beginning of the university period has on the mental health of the students [15,28,29]”. As far as I understand, in this study, only financial family support has been analyzed, so the suggestions should accommodate this question.

In general, the implications of the results should be analyzed in greater depth.

Likewise, more study limitations should be indicated (e.g., cross-sectional study, use of self-report procedures, etc.).

RE: We apologize for the misunderstanding about the construct family support, including more than financial support. In addition, we have corrected it and we have deeply analysed all the implications and limitations of the study, mainly those related to the family support, and we have elaborated in a greater extent the limitations section.

Changes in text:

Pease see introduction and discussion section.

Reviewer 2 Report

ijerph-456395_v1: “Psychological Distress, Family Support and Employment Status in University Students of Spain”

In this study, more than 3,700 first-year university students from Spain were analyzed in regard to psychological distress, family support and employment status of the student. More female than male students showed symptoms of psychological distress. Results further indicate that less family support was associated with greater rates of psychological distress and that female students who were studying and looking for work showed greater psychological distress compared to female students who were only studying or who were studying and also working.

Analyzing psychological distress during transition to university is an interesting approach and thus this study is not without merit. However, I have several concerns (in particular, in regard to methodology) as outlined below.

1)    Introduction:

- The authors introduce their research topic fairly superficial. I miss definitions of what they mean when they talk about ‘psychological distress’ or ‘family support’. Further they should provide some examples of the studies they cite (e.g., p.2, l.68: psychological distress varies depending on specific samples).

- The introduction is fairly short and authors should consider introducing the specific (Spanish) context of their study (e.g., provide the percentage of young adults who study in Spain, economic situation, etc.). Further, I wonder why they analyze student employment status and do not consider the financial resources students have at hand (which probably will be more important and the reason why some students need/do not need to work).

- p.2,l.73: ‘it’ instead of ‘is’

2)    Methods

- The overall response rate extremely low. What do the authors know about the students who decided to participate and how do they differentiate from students who did not.

- The authors need to provide measures of reliability for all of their measures and they should describe their measurement instruments in much more detail and provide example items. Further, for self-perceived health status, authors should provide the numbers of all categories

- Why did the authors use a categorial approach in regard to ‘psychological distress’ (yes/no) instead of using the complete scale from 0 to 12 (dimensional approach in which more of the available information will be used)?

- The section on data analysis is not easy to follow and it is not clear how the authors did the multivariate analysis (in Table 3 only some variables are included, however authors write that the models were adjusted for other variables). Much more information needs to be provided.

- p.4, l.158ff. : Authors left editorial instructions in the text – please delete.

3)    Results

- Given their relatively large sample, authors need to provide effect sizes in addition to p-values (e.g. Cramers-V for the distribution of variables)

- Authors need to provide more explanations/information on their tables.

4)    Discussion

- The authors discuss the different prevalence of ‘psychological distress’ in their sample compared to former studies. However, they miss to mention the self-selection bias in their own sample (only 4.6% of students participated).

- The authors correctly point out the possible bi-directional relationship (p.11,l.233). However, they still over-interpret their findings later on and do not discuss the limitation that no causal implications can be drawn from their correlational study.

- p.11,l.253: delete “might” – the response rate certainly has an impact on the findings.

- Authors do not discuss the role of family support in enough detail. Further, they can only speculate about their results in regard to employment status as they did no assess financial resources.

A more general comment:

The manuscript reads generally well, however, sometimes the wording is a bit awkward. I recommend that an English native speaker proofreads the text.

Author Response

First of all, we would like to thank the reviewers for the effort and time employed in reviewing our article. We found the comments and suggestions pertinent and are convinced they have helped us improve our paper considerably.

We have responded to all of the comments one by one here below, and have adjusted the main document accordingly.

1)    Introduction:

- The authors introduce their research topic fairly superficial. I miss definitions of what they mean when they talk about ‘psychological distress’ or ‘family support’. Further they should provide some examples of the studies they cite (e.g., p.2, l.68: psychological distress varies depending on specific samples).

RE: We agree with the reviewer in the superficiality in the explanation of the psychological distress and family support concepts. Thus, more information regarding these concepts was provided and some examples of the specific studies cited were provided.

Changes in text:

Introduction, please see 55 to 58 and 74 to 81

- The introduction is fairly short and authors should consider introducing the specific (Spanish) context of their study (e.g., provide the percentage of young adults who study in Spain, economic situation, etc.). Further, I wonder why they analyze student employment status and do not consider the financial resources students have at hand (which probably will be more important and the reason why some students need/do not need to work).

RE: We greatly appreciate this comment. We have now elaborated in a greater extent the introduction. In this sense, we have extended the whole introduction to give a wider focus of our work adding some details of the specific context.

Changes in text:

Please see highlighted changes in the introduction.

- p.2,l.73: ‘it’ instead of ‘is’

 RE: We apologize for the mistake. Consequently, we changed it for is.

Changes in text:

Introduction, please see line 84

2)    Methods

- The overall response rate extremely low. What do the authors know about the students who decided to participate and how do they differentiate from students who did not.

RE: We added a sentence specifying the degree of precedence of the students and the three most common degrees in which the students were enrolled.

Changes in text:

Methods please see lines 109 to 111

- The authors need to provide measures of reliability for all of their measures and they should describe their measurement instruments in much more detail and provide example items. Further, for self-perceived health status, authors should provide the numbers of all categories

RE: We appreciate this comment and consequently we added a deeper description of the tools used in the study. Regarding self-perceived health status, due to limitations related to the possible reversal causality, was included mainly as adjustment and not as one of the main independent factors. Due to this and after cautiously consideration, we have decided to maintain this variable as a dichotomous one. In addition, some categories had a low number of individuals, reason for what we suppose that pooling their punctuations could provide more robust measures.

Changes in text:

Methods please see study variables definitions

- Why did the authors use a categorial approach in regard to ‘psychological distress’ (yes/no) instead of using the complete scale from 0 to 12 (dimensional approach in which more of the available information will be used)?

RE: We acknowledge this comment and agree with it in sense that we can loss information. Nevertheless, to consider the main outcome as a dichotomous variable was based on the proposal of the author of the scale. In addition, previous studies have used this approach, facilitating the comparison between results. Due to this, we have decided to maintain this variable as a dichotomous one.

Changes in text:

Methods, please see line 142 and 143

- The section on data analysis is not easy to follow and it is not clear how the authors did the multivariate analysis (in Table 3 only some variables are included, however authors write that the models were adjusted for other variables). Much more information needs to be provided.

RE: We appreciate this comment and its consequences in the reproducibility. Consequently, it was included more information about how the multivariate analysis was performed.

Changes in text:

Methods, please see line 183 to 185

- p.4, l.158ff. : Authors left editorial instructions in the text – please delete.

 RE: We apologize for the mistake. These instructions were taken out.

3)    Results

- Given their relatively large sample, authors need to provide effect sizes in addition to p-values (e.g. Cramers-V for the distribution of variables)

RE: We completely agree with this appreciation about the p values and the influence on them of the sample size. However, given that the main results of our study are those from the regression models and the Odds Ratio were provided with their confidence intervals, we think that it might be enough to have an idea about how the sample size could influence the estimations. Regarding the distribution of variables in the sample, its main objective it is not to see their effect on the main outcome. For this reason and after cautiously and deeply consider it, we have decided not to include the effect sizes in table one.

Changes in text:

Results, please see tables

- Authors need to provide more explanations/information on their tables.

RE: More information was included in tables, particularly about how the p values for bivariate analyses were obtained, how the multivariate analysis was performed and a specification about the goodness of fit of the models

Changes in text:

Results, please see tables

4)    Discussion

- The authors discuss the different prevalence of ‘psychological distress’ in their sample compared to former studies. However, they miss to mention the self-selection bias in their own sample (only 4.6% of students participated).

Re: We have now included explicitly the self-selection bias as a limitation of the study.

Changes in text:

Discussion, please see limitations

- The authors correctly point out the possible bi-directional relationship (p.11,l.233). However, they still over-interpret their findings later on and do not discuss the limitation that no causal implications can be drawn from their correlational study.

Re: The fact that no causal implications can be drawn from our study was explicitly included in discussion and the limitations section was included

Changes in text:

Discussion, please see limitations.

- p.11,l.253: delete “might” – the response rate certainly has an impact on the findings.

Re: We apologize for the mistake and deleted “might”.

Changes in text:

Discussion, please see limitations.

- Authors do not discuss the role of family support in enough detail. Further, they can only speculate about their results in regard to employment status as they did no assess financial resources.

 Re: A deeper discussion about the role of family support was elaborated.

Changes in text:

Discussion, please see 255 to 260

A more general comment:

The manuscript reads generally well, however, sometimes the wording is a bit awkward. I recommend that an English native speaker proofreads the text.

RE: The manuscript was reviewed by a English native speaker.

We hope these changes meet your expectancies and we would like to thank you again for the effort and time spent reviewing our article.

Reviewer 3 Report

You missed taking into account a wide literature on academic students referring to academic burnout. Is this different from stress that you used? I'm sure it could add a new perspective in your theoretical framework. As an example you may consider the following scholars' work:

Salmela-Aro, K.; Fiorilli, C.,

With regard to the Spanish context:

Durán, M. A., Extremera, N., Rey, L., Fernández-Berrocal, P., & Montalbán, F. M. (2006). Predicting academic burnout and engagement in educational settings: Assessing the incremental validity of perceived emotional intelligence beyond perceived stress and general self-efficacy. Psicothema,18.

Merino-Tejedor, E., Hontangas, P. M., & Boada-Grau, J. (2016). Career adaptability and its relation to self-regulation, career construction, and academic engagement among Spanish university students. Journal of Vocational Behavior,93, 92-102.

Why you collected data addressing students' sexual orientation: Are there relevance in the existing literature?

Method:

Do you think that the family APGAR developed in 1978 by Smilkstein is still a good instrument for modern society? Could you quote recent adoptions of the instrument?

Results and discussions

You compared your results with others coming from the previous study which collected data by different instruments. You should highlight that it could be a weak comparison. You couldn't stress the idea that your first-year University students are more at risk without adding some limits: the absence of the same instruments to compare cross-countries data; a longitudinal study design which could better explain your suppositions

Author Response

Reviewer #3

You missed taking into account a wide literature on academic students referring to academic burnout. Is this different from stress that you used? I'm sure it could add a new perspective in your theoretical framework. As an example you may consider the following scholars' work:

Salmela-Aro, K.; Fiorilli, C.,

With regard to the Spanish context:

Durán, M. A., Extremera, N., Rey, L., Fernández-Berrocal, P., & Montalbán, F. M. (2006). Predicting academic burnout and engagement in educational settings: Assessing the incremental validity of perceived emotional intelligence beyond perceived stress and general self-efficacy.Psicothema,18.

Merino-Tejedor, E., Hontangas, P. M., & Boada-Grau, J. (2016). Career adaptability and its relation to self-regulation, career construction, and academic engagement among Spanish university students. Journal of Vocational Behavior,93, 92-102.

Why you collected data addressing students' sexual orientation: Are there relevance in the existing literature?

RE: We thank o the reviewer for these comments, the references and the names of important authors on the topic provided. In this sense, we have added some information in the introduction, and some references both in the introduction and in the discussion. Furthermore and about the sexual orientation, we want to explain here that this variable was included mainly as adjustment factor due to the evidence found in previous research, such as the studies listed here below among others.

1: Ueno K, Vaghela P, Nix AN. Gender composition of the occupation, sexual orientation, and mental health in young adulthood. Stress Health. 2018 Feb;34(1):3-14. doi: 10.1002/smi.2755. Epub 2017 Apr 18. PubMed PMID: 28417545.

2: Lourie MA, Needham BL. Sexual Orientation Discordance and Young Adult Mental Health. J Youth Adolesc. 2017 May;46(5):943-954. doi: 10.1007/s10964-016-0553-8.  Epub 2016 Aug 1. PubMed PMID: 27480273.

3: Rodriguez-Seijas C, Eaton NR, Pachankis JE. Prevalence of psychiatric disorders at the intersection of race and sexual orientation: Results from the National Epidemiologic Survey of Alcohol and Related Conditions-III. J Consult Clin Psychol. 2019 Apr;87(4):321-331. doi: 10.1037/ccp0000377. PubMed PMID: 30883161.

Changes in text:

Please see introduction (line 96)

Please see discusion (lines 261 to 263)

Please see references

Method:

Do you think that the family APGAR developed in 1978 by Smilkstein is still a good instrument for modern society? Could you quote recent adoptions of the instrument?

RE: While the metric properties Family APGAR possibly should be reviewed due to social changes, particularly during last years in the context of the economic recession, it is still used by many studies. In these sense we have added in the manuscript a brief explanation of the fact that is still in use and some references to support this affirmation.

Changes in text:

Please see methods (lines 156)

Please see references

Results and discussions

You compared your results with others coming from the previous study which collected data by different instruments. You should highlight that it could be a weak comparison. You couldn't stress the idea that your first-year University students are more at risk without adding some limits: the absence of the same instruments to compare cross-countries data; a longitudinal study design which could better explain your suppositions

RE: We want to give thanks again to the reviewer for the comments done and particularly for this one. In this sense, we have included more information in the limitations section.

Changes in text:

Please see discussion (lines 240 to 242 and 292 to 294)

Reviewer 4 Report

The paper “Psychological Distress, Family Support and Employment Status in First-Year University Students of Spain” addresses the aims and scopes of the journal and makes a contribution to the literature on the topic analyzed.
The title clearly describes the article, the abstract reflects the content of the article, the authors include salient key words.
The figures and tables are relevant to the discussion in the text.

Minor Revisions

I find the meaning unclear.
Please, verify:
- page 1, lines 40 and 41: (49.9% + 54.2%) = 101.1% ; It should be 100% . Please, correct.
- page 3, line 106 and lines 124-129: It is written "The study population consists of 4,166 first-year university students" ... "the following participants were excluded: ..." ..." In total, a sample of 3,717 students"; but 4,166-(314+4+13+41+86)=3,708. Also in Tables the total sample corresponds to 3,717. Please, correct.

English
Please, check it:
- page 2, line 89: I suggest to delete "both”.
- page 3, line 120: It is written "allows keep the confidentiality of data and thus comply",
I suggest "allows to keep the confidentiality of data and thus to comply".

Author Response

Reviewer #4

The paper “Psychological Distress, Family Support and Employment Status in First-Year University Students of Spain” addresses the aims and scopes of the journal and makes a contribution to the literature on the topic analyzed.

The title clearly describes the article, the abstract reflects the content of the article, the authors include salient key words.

The figures and tables are relevant to the discussion in the text.

RE: We want to give thanks the reviewer for the effort made and the time spent reviewing our article. About the revision, again thank you for consider that our paper makes a contribution to the topic. Here below we response to one-by-one to the comments done. Again thank you for the improvement in the quality of the article that the changes derived from these comments could produce on it.

Minor Revisions

I find the meaning unclear.

Please, verify:

- page 1, lines 40 and 41: (49.9% + 54.2%) = 101.1% ; It should be 100% . Please, correct.

RE: Thank you for this comment. About it, we should explain that the prevalence was calculate separately form men and women, due to this the sum could be higher or lower than 100%. In this sense, we have clarified the sentence.

Changes in text:

Please see abstract (lines 40 to 42)

- page 3, line 106 and lines 124-129: It is written "The study population consists of 4,166 first-year university students" ... "the following participants were excluded: ..." ..." In total, a sample of 3,717 students"; but 4,166-(314+4+13+41+86)=3,708. Also in Tables the total sample corresponds to 3,717. Please, correct.

RE: We strongly agree with the comment of the author about the mismatch of numbers. In this sense, we have to comment that these numbers are correct, but we forgot to explain that some participants had missing values in more than one variable. Due to this, we have clarified it adding a sentence to explain this fact. Thank you.

Changes in text:

Please see methods (line 132)

English

Please, check it:

- page 2, line 89: I suggest to delete "both”.

- page 3, line 120: It is written "allows keep the confidentiality of data and thus comply",

I suggest "allows to keep the confidentiality of data and thus to comply".

RE: Apologies for the mistakes and many thanks again for the comments. We have changed it.

Changes in text:

Please see introduction (line 91) and methods (lines 122 and 123)

Round 2

Reviewer 1 Report

In my opinion, the authors have responded satisfactorily to all the questions requested, so the manuscript meets the requirements for publication in its current version. However, I would recommend a small improvement in the Introduction section. Although the authors adequately justify their hypothesis about the positive relationship they expect to find between employment and psychological distress, in my view, they do not specify in which of the two genders they expect this relationship to be higher. Based on the information provided previously, it can be inferred that they expect to find a significantly higher relationship in women than in men, but they should expressly indicate this (and why). 

Author Response

Reviewer #1

In my opinion, the authors have responded satisfactorily to all the questions requested, so the manuscript meets the requirements for publication in its current version. However, I would recommend a small improvement in the Introduction section. Although the authors adequately justify their hypothesis about the positive relationship they expect to find between employment and psychological distress, in my view, they do not specify in which of the two genders they expect this relationship to be higher. Based on the information provided previously, it can be inferred that they expect to find a significantly higher relationship in women than in men, but they should expressly indicate this (and why).

RE: We would like to thank you again for the effort and time spent reviewing our article. Sincerely, thank you. In addition, we appreciate your generosity in considering our work for publication. Regarding the comment about in which gender is expected a stronger relationship between the psychological distress and employment status, some information and a reference about it was added in the introduction.

Changes in text:

Please see introduction (lines 95 to 97)

Reviewer 2 Report

ijerph-456395_v2: “Psychological Distress, Family Support and Employment Status in First-Year University Students of Spain”

In their revision, the authors were unfortunately not able to solve the most important issues of the first manuscript version:

-        Instead of defining the key constructs (i.e., psychological distress and family support) and providing examples for the varying prevalence of psychological distress in students in other students as suggested, they just rephrased and moved some content (e.g., p.2, l. 65: still no examples of these studies are provided; p. 2, l. 68ff. is just a repetition of the first paragraph, etc.). Further, still no information on the Spanish context are provided.

-        While the authors now provide information about the three groups of students which add to a fifth of the sample, we still do not know, why these students decided to participate and why 95.4% of the invited students did not and how these groups compare

-        While it may be ok to differentiate between good and poor health status, the authors still should provide the numbers of all categories in Tab.1.

-        The section on data analysis is still not easy to follow and it is still not clear how the authors did the multivariate analysis (e.g., what is the meaning of “…using a backward stepwise method with significance level of 0.1 for removal from the model of 0.05 for addition to the model” (p.4, l.184f.) – should there be an “and” instead of “of”?).

-        I understand that the distribution of variables in the sample is not the main focus of this study. However, authors still should consider providing Cramer V as an indicator (otherwise the comparison of men and women in table 1 would be hard to interpret).

-        Still not much information about the tables and how to interpret them is provided by the authors

-        The authors now discuss their limitations in a much better way, however, they still do not discuss the role and possible mechanisms at work of family support and employment status in enough detail.

-        I still recommend that an English native speaker proofreads the text.

Author Response

Reviewer #2

In their revision, the authors were unfortunately not able to solve the most important issues of the first manuscript version:

Instead of defining the key constructs (i.e., psychological distress and family support) and providing examples for the varying prevalence of psychological distress in students in other students as suggested, they just rephrased and moved some content (e.g., p.2, l. 65: still no examples of these studies are provided; p. 2, l. 68ff. is just a repetition of the first paragraph, etc.). Further, still no information on the Spanish context are provided.

RE: We have added some information about the Spanish context and include some references to support this information.

While the authors now provide information about the three groups of students which add to a fifth of the sample, we still do not know, why these students decided to participate and why 95.4% of the invited students did not and how these groups compare

RE: We thank the reviewer for this appreciation. About it we want to point out that some information about students was added in the previous revision. In addition a reference of the cohort composition was included, reason for what we think that it could be enough to understand the sample composition.

While it may be ok to differentiate between good and poor health status, the authors still should provide the numbers of all categories in Tab.1.

RE: We thank the reviewer again for this comment. As we explained in the previous revision, this variable was considered mainly for adjustment. In addition previous research have used this variable as a dichotomous one. Also, we want to point out that the estimates from models with the variable including more levels, does not change significantly the estimates for example comparing their results using a likelihood ratio test.

The section on data analysis is still not easy to follow and it is still not clear how the authors did the multivariate analysis (e.g., what is the meaning of “…using a backward stepwise method with significance level of 0.1 for removal from the model of 0.05 for addition to the model” (p.4, l.184f.) – should there be an “and” instead of “of”?).

RE: We apologize for the mistake and accordingly we have changed “of” by “and”

I understand that the distribution of variables in the sample is not the main focus of this study. However, authors still should consider providing Cramer V as an indicator (otherwise the comparison of men and women in table 1 would be hard to interpret).

RE: We thank the reviewer again for this comment. In this sense, we want to highlight that the p values of Table 1 are a comparison of patients characteristics and that Cramers V is an indicator of effect size. Also, we think that most of investigators are much more accustomed to use chi square tests are than Cramers V tests, reason for what we think that it could be easiest of understand for them.

Still not much information about the tables and how to interpret them is provided by the authors

RE: Again we apologize for any lack of information. In this sense, in the previous review we have added some specifications as footnotes.

The authors now discuss their limitations in a much better way, however, they still do not discuss the role and possible mechanisms at work of family support and employment status in enough detail.

RE: We thank the reviewer for this comment. In this sense we want to highlight that some aspects of these relations were added during the review process.

I still recommend that an English native speaker proofreads the text.

RE: We apologize for the mistakes. The English was reviewed by a native speaker